# Screening of Cognitive Changes in Adults with Intellectual Disabilities: A Systematic Review

**DOI:** 10.3390/brainsci10110848

**Published:** 2020-11-12

**Authors:** Andreia F. Paiva, Adam Nolan, Charlotte Thumser, Flávia H. Santos

**Affiliations:** 1Communication and Society Research Centre, University of Minho, 4710-057 Braga, Portugal; andreiafonsecapaiva@gmail.com; 2Centre for Disability Studies, University College Dublin Belfield, Dublin 4, Ireland; adam.nolan@ucdconnect.ie; 3College of Social Sciences and Law, University College Dublin, Belfield, Dublin 4, Ireland; charlotte.thumser@ucdconnect.ie; 4School of Psychology, University College Dublin, Belfield, Dublin 4, Ireland

**Keywords:** screening, dementia, intellectual disability, early-onset, neuropsychology

## Abstract

Background and Aims: Screening and assessment of cognitive changes in adults with Intellectual Disabilities (ID), mainly Down Syndrome (DS), is crucial to offer appropriate services to their needs. We present a systematic review of the existing instruments assessing dementia, aiming to support researchers and clinicians’ best practice. Methods: Searches were carried out in the databases Web of Science; PubMed; PsycINFO in March 2019 and updated in October 2020. Studies were selected and examined if they: (1) focused on assessing age-related cognitive changes in persons with ID; (2) included adults and/or older adults; (3) included scales and batteries for cognitive assessment. Results: Forty-eight cross-sectional studies and twenty-seven longitudinal studies were selected representing a total sample of 6451 participants (4650 DS and 1801 with other ID). In those studies, we found 39 scales, questionnaires, and inventories, and 13 batteries for assessing cognitive and behavioural changes in adults with DS and other ID. Conclusion: The most used instrument completed by an informant or carer was the Dementia Questionnaire for Learning Disabilities (DLD), and its previous versions. We discuss the strengths and limitations of the instruments and outline recommendations for future use.

## 1. Introduction

Individuals with intellectual disabilities (ID) may be at an increased risk of developing dementia when compared to the general population [1]. In people with ID, the prevalence of dementia is as high as 4% in individuals under 40 years, and 40% in those 60 years or older, with an average age of onset between 51 and 56 years [2,3,4]. Epidemiological studies found that within a population of 222 individuals with ID aged 60 years, a total of 29 had a dementia diagnosis when using the criteria from both the International Classification of Diseases, 10th Revision (ICD-10) and Diagnostic and Statistical Manual of Mental Disorders, 5th Edition (DSM-5) [5]. Among those diagnosed with dementia, 66% of individuals met criteria for dementia of Alzheimer’s type, with a prevalence of 8.6% (95% CI 5.2–13.0). Recently, a cross-sectional study with 493 adults with Down Syndrome (DS) and other ID reported that individuals with other ID may develop dementia and mild neurocognitive disorder at an earlier age and at a higher rate than the general population. The prevalence of dementia in individuals with other ID was 0.8% in the age group of 45 to 54 years, 3.5% in the group of 55 to 64 years and 13.9% for those aged 65 to 74 years. The study also showed that the prevalence of mild neurocognitive disorder in individuals with other ID was 3.1% in the age group of 45 to 54, 3.5% in the age group of 55 to 64, and 2.8% in the age group of 65 to 74. When analysed by severity of ID in individuals with DS and other ID, 1.5% of the individuals with moderate ID were diagnosed with dementia, 5.0% with severe ID were diagnosed with dementia in relation to 3.0% of individuals with moderate ID and 1.7% with severe ID were diagnosed with mild neurocognitive disorder [6].

Pathological studies also provide evidence for early-onset dementia. One study reported that by the age of 40 years, nearly all individuals with DS presented Alzheimer’s Disease (AD) markers [7], while longitudinal studies show that by the age of 65 years over 90% of people with DS and other ID meet diagnostic criteria for dementia [8,9]. Another study carried out with individuals with DS and other ID (*n* = 526) showed that among individuals with a diagnosis of DS, symptoms of dementia appeared earlier than those in other ID (average age of diagnosis was 52 years of age). In 75% of the cases, the symptoms were consistent with dementia of Alzheimer’s type [10].

Early detection of dementia can be challenging in individuals with ID [11]; many of the instruments for assessing dementia-related cognitive changes in the general population are based on the assumption of sound premorbid cognitive functioning, which is difficult to determine in those with ID [12,13,14]. Furthermore, the clinical presentation of dementia in those with ID may differ compared to the general population, with personality and behavioural changes presenting earlier [15,16].

Single domain cognitive tests are the usual approach to screen for dementia in the general population, as they can identify progressive deterioration in cognitive domains [17]. However, in people with ID, these tests are not appropriate due to pre-existing conditions which makes it difficult to determine baseline cognitive function, meaning the results cannot be interpreted in a substantial and valid way, as there are often no norms for this population [11]. This has been addressed in recent research carried out by Benejam [18], who used the CAMCOG-DS in people with Down syndrome to accurately diagnose Alzheimer’s disease. This shows the importance of developing reliable population norms for appropriate instruments when assessing cognitive changes in people with ID.

### 1.1. Down Syndrome Intellectual Disability

Among adults with ID, there is a well-established link between DS and dementia, particularly AD. Research indicates that 95% of people with DS will develop AD by the age of 65 [4,19,20]. Individuals with DS also have an increased risk of developing early-onset dementia; the clinical presentation of dementia symptoms before the age of 65 [4,19,21]. The increased prevalence of AD in DS is largely due to genetic factors associated with trisomy 21, the most common form of DS. Those with trisomy 21 have a third copy of chromosome 21 [22], which is responsible for the production of β-amyloid precursor protein [23]. The increased presence of β-amyloid precursor protein leads to an accelerated build-up of senile plaque in the brain, which is a primary cause of AD [22]. By age 40, most individuals with DS display neuropathological changes consistent with AD, while most individuals with DS show clinical signs of dementia by age 50 [24]. Similarities of symptoms between AD and DS suggest common risk factors among AD and DS. Prasher and colleagues (2008) [25] examined Apolipoprotein (APOE) genotyping in people with DS, concluding that those with APOE E4 allele had a significantly higher risk of developing AD, had an earlier onset of AD, and a higher rate of progression to death when comparing for participants with APOE 3 allele. Screening for APOE genotype in this population may be of good clinical utility as it helps people obtain early treatment, which can reduce early mortality rates [25,26]. Startin et al. (2019) [27] recently “conducted the largest cognitive study to date” (p. 245) with 312 participants with DS in order to assess typical age-related and AD-related cognitive changes in this population. The authors reported memory and attention measures were most sensitive to decline, although the earliest cognitive markers of AD-related pathology were identified on most outcome measures. They also reported an age-related relationship where older age groups showed poorer performance in neuropsychological tests, except for scores on the Behaviour Rating Inventory of Executive Function—adult version; a measure of executive function. However, other research has indicated that declines in executive function may precede memory loss in those with DS and AD [28], suggesting further research is needed to determine the typical progression of AD in this population.

### 1.2. Other Intellectual Disability

There is less conclusive evidence of an increased risk of dementia in those with an intellectual disability not related to DS (herein other ID). While there may be several genetic factors, leading to increased risk of dementia in those with other ID—such as reduced baseline cognitive ability and fewer neurons and synaptic connections [1]—older adults with other ID show protective factors against developing dementia, including lower rates of smoking and greater cardiovascular health compared to the general population [29].

Some research suggests the prevalence of dementia for individuals with other ID may be the same or slightly higher than the general population [30,31], although a longitudinal study by Strydom et al. (2013) [1] reported that dementia might be five times more prevalent in this population. However, epidemiological studies may underestimate true prevalence rates due to several factors. Firstly, dementia is under-diagnosed in the general population—it is likely that this is also present in those with ID [14]. Secondly, those with ID generally have poorer access to health care services [32,33], which could result in lower levels of diagnosis. Finally, dementia presents differently in those with ID compared to those without, leading to difficulty in diagnosis [14].

### 1.3. The Present Study

Due to the prevalence of dementia in those with ID, particularly DS, it is important that researchers and clinicians have validated, reliable measures for diagnosis. Standardised measures are necessary for determining prevalence within a population, assessing and comparing interventions, and synthesising research findings for meta-analyses; however, a systematic review by Zellinger et al. (2013) [14] noted an “immense” number of instruments assessing cognitive change in those with ID. The present review aims to build on the previous work by Zellinger et al. (2013) [14] by comprehensively reviewing the existing instruments available for screening for cognitive impairments in individuals with ID, considering cross-sectional and longitudinal studies. This systematic review focuses on scales and batteries as they demonstrate a more robust way to screen for dementia in this population [14,17]. The review will look at the strengths and limitations of instruments and aims to provide researchers and clinicians with an up to date, comprehensive list of available tools.

## 2. Materials and Methods

The methods for this review were based on the recommendations of the Preferred Reporting Items for Systematic Reviews and Meta-Analyses [34]. As a complement and extension to the PRISMA protocol, we used the Synthesis Without Meta-Analysis in Systematic Reviews Checklist (SWiM), following the recommendation of the EQUATOR group network (“Enhancing the QUAlity and Transparency Of Health Research”) (as seen in https://www.equator-network.org) [35]. Both checklists, quality assessment and eligible studies, are available as Appendix A.

### 2.1. Literature Search

Two systematic literature searches of three databases (Web of Science; PubMed; PsycINFO) were conducted. Searches included the key terms (with the appropriate Boolean operators for each database) “Adult* OR Older adult*”; “Cognit* task OR Cognit* test OR neuropsych* test”; “Instrument* OR Scale OR questionnaire OR screening”; “Dementia”; “Intellectual* Disabilit* OR mental* retar* OR General learn* disabilit*”. Filters were applied for the key terms NOT “Child* AND adolesc* AND youth*”. Searches were performed with consideration of all articles, without limiting the year of publication or language of publication. Except for two publications, one in Spanish and one in German, both included in the screening phase, all other search results were published in English.

### 2.2. Eligibility Criteria and Data Extraction

The eligibility criteria for the studies included in this systematic review were:

Population: Studies that included adults aged 18 years and older diagnosed with Intellectual Disability;

Intervention: Screening of cognitive changes in adults with Intellectual Disabilities;

Comparators: Studies using scales and batteries to assess cognitive changes and dementia in individuals with intellectual disabilities including Down Syndrome;

Outcomes: Studies assessing cognitive and behavioural changes in adults with intellectual disabilities;

Studies: Studies with cross-sectional and longitudinal designs.

During the first search in March 2019, 70 articles were found on Web of Science, 76 on PubMed, and 60 on PsycINFO (*n* = 206). Duplicated records (*n* = 63) were removed, leaving 143 articles. A second search for new entries to databases using the same key search terms was done in September 2019 and 58 new entries were found. The search was repeated in May of 2020 and no new articles were identified, and one article was added in October 2020.

All 202 titles and abstracts were screened using the following inclusion criteria: (1) studies focusing on assessment of dementia in person with ID; (2) population being adults and/or older adults; (3) studies including scales and batteries for cognitive assessment. Sixty-one articles were excluded based on exclusion criteria (review studies and/or intervention studies, or the age of participants not matching the criteria). In total, 140 articles were included for a thorough review (as shown in Figure 1). A manual search of the reference sections of the retrieved studies and review articles was conducted. However, no new articles meeting the inclusion criteria were found.

We analysed 48 cross-sectional studies and 27 longitudinal studies qualitatively, excluding 66 articles for not meeting inclusion criteria (e.g., review studies, intervention studies, and studies including children or adolescents). In total, 75 articles were included in this review. All articles were reviewed by two researchers independently. In the few cases of disagreement, discrepancies were solved by consensus.

### 2.3. Quality Assessment

As for critical appraisal of the studies included in this review, a standardised checklist to identify the risk of bias was used to assess the quality of included studies. The checklist was based on the Newcastle–Ottawa Scale (NOS) [36], embedded on the Table A2 and Table A3. A total score with a maximum value of nine points provides a rating for the quality level. Quality levels of evidence were defined as high (9–7 points); medium (6–4 points), and low (3–1 point). No studies presented low-quality range.

## 3. Results

### Descriptive Synthesis

This review identified 48 cross-sectional studies and 27 longitudinal studies with ID population testing. Cross-sectional studies were conducted in the United Kingdom (13), United States (17), Spain (4), Netherlands (4), Italy (4), Ireland (2), Belgium and Switzerland (1), Australia (2) Israel (1), Finland (1) and Canada (1). Longitudinal studies were conducted in the United States (12), the United Kingdom (7), Ireland and the United States (1), Ireland (1), Germany (1), Canada (1), Australia (1), Spain (1), and the Netherlands (1). The most frequent journal in this review was the Journal of Intellectual Disability Research, with a H-index of 93 and an impact factor of 1.94.

Of the 48 cross-sectional studies, 24 included only participants with DS, while the remaining 24 included individuals with DS and other ID. Table 1 represents the demographic information for both cross-sectional and longitudinal studies. In longitudinal studies, the available *n* accounts for the average of individuals in the last wave (follow-up) of each study.

The tables for cross-sectional and longitudinal studies (Appendix B, Table A2 and Table A3) present the characteristics of the participants (age, diagnosis), intervention, comparison, outcomes, and study design [37] structured according to the eligibility criteria. The average duration of longitudinal studies was 97.01 months, with no data for one study. This was calculate based on the total amount of months for each study, from baseline to the last follow-up, dividing by the number of studies included (average = average + ((value − average)/nValues).

We found 39 scales, questionnaires, and inventories, and 13 batteries for assessing cognitive and behavioural changes in adults with ID (see Appendix C). A total of 23 informant-based measures (scales, questionnaires, and inventories) were used to obtain information on behavioural and cognitive changes from a proxy, while the remaining 29 instruments were self-report measures (13 batteries and 16 scales, questionnaires and inventories). Of the cross-sectional studies included, 15 studies used only self-report instruments, 10 studies used only informant-based instruments, and 15 studies used both type of instruments. Regarding the longitudinal studies, 10 studies used self-report instruments, 5 studies used only informant-based measures, and 7 studies used both types of measures. The remaining studies used single domain tests or tasks (8 cross-sectional studies, 5 longitudinal studies) (see Appendix B, Table A2 and Table A3). According to the selected studies, we identified a multitude of different instruments (single-domain cognitive tests; scales; batteries; tasks), with few replications, and a lack of descriptive data (means, standard deviations, gender ratios, specificity and sensitivity scores) in publishing material, which was not obtained from all authors upon request. Consequently, a meta-analysis could not be performed. Of the 27 longitudinal studies, the majority (*n* = 19) focused on DS, while the remainder (*n* = 8) included participants with DS and other ID. There was also a large degree of heterogeneity in measures used in longitudinal studies including those with both DS and other ID. Within the eight studies included, 30 measures and tasks were reported. All datasets generated for this study are included in the article or its Appendix A, including Appendix A list of instruments used in the studies, PRISMA checklist and SWiM checklist.

## 4. Discussion and Implications

This study aimed to systematically review scales and batteries for screening for cognitive changes in adults with ID and provide a guide for practitioners and researchers to choose valid, reliable instruments. This review found a multitude of materials used with adults with ID, with much of the research focusing on those with DS. We focused on batteries and scales as the best approach to evaluate cognitive changes and age-related changes in individuals with ID [14,17]. The current evidence encourages the focus on measures such as DLD and CAMCOG-DS, which should be further explored psychometrically, clinically and longitudinally among the essential clinical diagnosis tools to distinguish mild neurocognitive disorder and dementia status in those with ID, particularly DS [38].

Identified instruments can be divided into two categories: informant-based measures (answered by a carer) and self-report measures (answered by the individual). Across the literature, the diagnosis of dementia in this population is a major concern and subject to a disagreement regarding which instrument to use; there is also considerable disagreement surrounding which instruments better discriminate mild neurocognitive disorder and preclinical dementia [8]. Studies are discussed according to the study design and clinical groups.

### 4.1. Longitudinal Studies

#### 4.1.1. Longitudinal Studies in Participants with Down Syndrome

The present review identified a multitude of measures used to assess cognitive change in those with DS—36 separate measures and tasks were used across the 19 studies. The Dementia Questionnaire for Learning Difficulties (DLD—previously referred to as the Dementia Questionnaire for Persons with Mental Retardation, or DMR) [39,40,41] was the most frequently used measure, appearing in seven studies [4,8,38,42,43,44,45]. The frequent use of the DLD may reflect its recommendation by the National Institute for Health and Clinical Excellence—Social Care Institute for Excellence in the UK [46]. The DLD, an informant-based measure, was developed by Evenhuis (1990) [39] for use with Dutch speakers but has since been translated and used in several countries, allowing cross-cultural comparisons [10,43,45]. The DLD consists of 50 items and eight subscales and provides scores for cognitive and social domains. Previous research has noted that the DLD is widely used due to high levels of agreement between its scores and clinician’s diagnosis [47] as well as its good sensitivity and specificity [48].

In the included studies, the DLD was effective in identifying deterioration in cognitive and social skills in adults with DS over time [45], although Nelson et al. (2007) [44] noted that while DLD total scores showed good overall test-retest reliability after one year (*r* = 0.77), there was low test-retest reliability for the social scale (*r* = 0.45). In another study, [43], using the cognitive element of the DLD as a secondary measure to examine the impact of seizures on cognitive impairment in adults with DS, Lott et al. (2012) [43] found that the cognitive scale of the DLD identified increased deterioration in adults with DS and AD with seizures compared to those without seizures. Similarly, a 14-year longitudinal study by McCarron et al. (2014) [8] found that epilepsy was identified as a significant predictor of dementia in adults with DS and noted the DLD was the most sensitive instrument for tracking cognitive changes over time. However, another study [45] reported that the DLD showed poor sensitivity in distinguishing between dementia-related cognitive decline and depression, which is likely due to the inclusion of the social skills element of the questionnaire. Furthermore, Evenhuis et al. (2009) [40] suggested that this measure may not have adequate sensitivity when used with people with severe and/or profound ID due to a floor effect; similarly, it may also be problematic with those with mild ID due to a ceiling effect on cognitive function. A multi-wave study [38] found that the overall summary score of the DLD clearly identified individuals with mild neurocognitive disorder onset.

The Severe Impairment Battery (SIB) [49] is another measure of cognitive functioning which has been used longitudinally. The SIB is a self-report measure assessing cognitive function across nine domains: attention, language, orientation, memory, praxis, visuospatial perception, construction, social skills, and orientating head to name [50]. The SIB was used in four longitudinal studies exclusively examining those with DS [8,38,42,43]. Like the DLD, [43] the SIB was effective at tracking the cognitive decline in adults with DS and seizures; it was used as a secondary measure and provides a limited description of its effectiveness [8,42].

#### 4.1.2. Longitudinal Studies Including Participants with DS and Other ID

There was no overlap between measures used across studies, with no measure included in more than one study. This is illustrative of the lack of standardised measures for assessing cognitive decline in those with other ID and highlights the need for an accepted, recommended measure to allow synthesis across different studies.

It is interesting to note that the DLD was only used in a single study including participants with other ID [10]. The study found that the DLD showed good test-retest reliability within their sample and reported that DLD scores showed agreement with other measures of cognitive change used in their study.

One potentially promising new measure for assessing cognitive decline in those with other ID is the Wolfenbütteler Dementia Test for Individuals with Intellectual Disabilities (WDTIM). The WDTIM was used in a 2-year longitudinal study carried out by Kuske et al. (2017) [51] and was effective at detecting cognitive changes over time. The authors noted that the WDTIM was more effective when used in conjunction with the Dementia Screening Questionnaire for Individuals with Intellectual Disabilities (DSQIID) [52]—an informant-based measure. The combination of a self-report and informant-based measure could provide a useful method to cross-check screening. However, like the DLD, the WDTIM may be problematic when used with individuals with severe and/or profound ID [51].

### 4.2. Cross-Sectional Studies

#### 4.2.1. Cross-Sectional Studies in Participants with Down Syndrome

As was the case with longitudinal studies, the DLD [39] was the most frequently used instrument, appearing in eight studies [43,47,53,54,55,56,57]. While the DLD was generally reported as a good marker of cognitive decline and dementia in those with DS, [24], one study found no association between scores on the DLD and the presence of beta-amyloid precursor protein, a biological marker of senile plaques and neurofibrillary tangles present in AD. While this may indicate that the DLD lacks sensitivity in identifying early cognitive changes associated with AD in those with DS, the authors suggest that small sample size and lack of statistical power may have influenced their findings.

The SIB [49] was also frequently used, appearing in four cross-sectional studies [22,53,58,59]. Witts and Elder (1994) [59] carried out a preliminary study on the use of the SIB with adults with DS and concluded that the measure was suitable to assess cognitive function in this population. Furthermore, they noted that no floor or ceiling effects were observed in scores on the SIB—this is advantageous as it indicates that the measure can be used to assess cognitive function in a wide range of individuals with ID. A later study [53] reported that the SIB showed good concurrent validity with the DLD. However, unlike Witts and Elder (1994) [59], the authors reported evidence of ceiling effects, which has implications for the clinical usefulness of the measure [53]. They also identified the need for more longitudinal research to determine the effectiveness of the measure over time. Boada [60], using a between-groups design, observed greater impairment in the group with dementia and DS compared to individuals without dementia when using the DLD, but no difference between groups when using the SIB. According to the authors, the DLD is an appropriate functional instrument to assess for dementia in individuals with DS and other ID, while the SIB was not designed for the diagnosis of dementia of Alzheimer’s but rather as a measure to monitor cognitive decline in individuals with DS which offers objective function from a clinical view point. Another potential limitation of the SIB is reported by Head et al. (2011) [24], who noted that, like the DLD, there was no association between scores on the SIB and the presence of beta-amyloid precursor protein, which may indicate that the measure lacks sensitivity.

#### 4.2.2. Cross-Sectional Studies Including Participants with DS and Other ID

The DLD [39,40,41] revealed good psychometric properties in studies with participants with both DS as other ID. Eight studies used the DLD [47,61,62,63,64,65,66,67]. Shultz et al. (2004) [48] reported the sensitivity of the DLD as 0.65 and specificity 0.93. The instrument was found to be a good marker of the cognitive and affective symptoms observed in the early signs of dementia [65] and displays good inter-test validity with other instruments like the SIB [53] and the Alzheimer’s Functional Assessment Tool (AFAST) [63]. The DLD has shown adequate inter-rater reliability for all subscales, except behaviour and disturbance, with correlations of 0.68 or higher [40].

Due to problems with floor and ceiling effects in the assessment of people with ID, researchers have attempted to address this issue. Startin et al. (2016) [56] created a comprehensive neuropsychological assessment to evaluate people with DS and avoid ceiling and floor effects. The LonDownS Consortium identified a set of tests for the evaluation in people with DS with minimum floor and ceiling effects. The authors suggest that the battery is suitable for most adults with DS, although half the participants with both dementia and DS were unable to undertake any of the cognitive tasks in the battery, indicating that it may be useful for screening before the development of dementia [56].

Another measure was the Cambridge Cognitive Examination (CAMCOG). This was originally designed for use with the general population but was later adapted for the assessment of dementia in those with DS (CAMCOG-DS) [68]. Cross-sectional studies have shown that this instrument can reliably differentiate between older and younger participants, is useful when possible dementia is considered, and shows good internal reliability (Cochran’s alpha between 0.82–0.89 and test-retest reliability (*r* = 0.86) [69]. When comparing CAMCOG-DS scores in a sample of DS participants between 30 to 65 years old, a significant difference was found in the cognitive performance between younger participants (30–44 years old) and older participants (>45 years old), except on the Attention/Calculation subscales [68]. This is consistent with the idea that the largest differences between age groups are in memory, praxis, and perception subscales [69,70]. The authors found a good correlation between MMSE and CAMCOG-DS scores (*r* = 0.97). This inter-test reliability remained after removing MMSE related items in the CAMCOG-DS and excluding participants who achieved zero scores (*r* = 0.95). Furthermore, recent research has identified recommended cut-off points for the CAMCOG based on a normative sample of adults with DS [18]. However, it has been noted that this measure may not be suitable for those with severe learning disabilities, severe sensory impairments, or advanced dementia due to floor effects [69]. This instrument has also been found to have “limited diagnostic value as a single assessment” because it is not possible to estimate the extent of the decline in cognitive functioning based on scores—the instrument is also limited at determining whether cognitive decline is due to ID, dementia, or other reasons [67].

There is evidence that the Test for Severe Impairment (TSI) is reliable for monitoring the progression of dementia in people with severe ID [71]. The TSI was used in three of the cross-sectional studies including participants with DS and other ID [61,72,73]. This instrument was developed to assess cognition in people with severe cognitive impairment, and most individuals with moderate/severe ID score on this test and only those with advanced dementia fail to score. In addition to its use in cross-sectional studies, the TSI is reliable and valid in longitudinal studies as it monitors rates of changes and indicates a decline in cognitive function over time that can indicate dementia. In one of the earliest studies using the TSI, [71], the authors assessed the reliability and validity of the instruments in a sample of 60 adults with DS. They found that the convergent validity of the TSI for all samples was good (*r* = 0.94), with satisfactory interrater reliability (*r* = 0.97) and test-retest reliability (*r* = 0.98) over a two-year period. The instrument also showed good internal consistency, with a Cronbach’s alpha of 0.89.

Although DLD has been used in most studies showing it to be effective in identifying changes over time in people with DS and other ID [45], one study [51] revealed that it may not be an appropriate measure to assess dementia in people with severe ID. Recently, DLD was used in Benejam [18] and as expected, participants with ID with prodromal AD and AD dementia had worse scores than asymptomatic subjects. These authors also recommend cut-off points for the CAMCOG-DS for a diagnosis of prodromal AD and AD dementia in adults with DS, based on population norms stratified by level of ID impairment: mild ID, a score of 80 and moderate ID, scores of 56.

When screening for cognitive decline in people with ID, we need to highlight and concentrate on the change and decline based on premorbid level of functioning [74]. It is important to keep in mind the ceiling effects of some measures in individuals with DS when compared to severe ID, for example of the SBI, which has implications for the clinical usefulness of the measure [59,64]. When using the same instrument on individuals with DS when compared to other ID, the TSI can be used in both DS and other ID due to the absence of ceiling and floor effects in individuals with moderate and severe ID, it is a valid and reliable measure to both DS and other ID [71,74].

#### 4.2.3. Other Measures

Across most studies, the findings suggest that people with ID performed more poorly in verbal tasks, with significant declines with age [61,75,76,77]. Phonological tasks are more likely to be sensitive to the detection of cognitive decline among individuals with DS compared to those with other ID, based on significant declines in these tasks [75,78]. This is an important finding when considering which assessment should be used for those with DS and those with other ID.

According to ICD-11 (World Health Organization/2019) and DSM-5 (American Psychiatric Association/2013), the diagnosis of dementia and cognitive changes in the general population and people with ID requires multi domain assessment. Thus, this finding means that phonological tasks are a cognitive marker that should be part of any protocol rather than be taken in isolation [17,18].

Another important aspect of the screening instruments for dementia in ID is their ability to assess the behaviour changes commonly seen during the onset of dementia. An example of this concern is the Assessment for Adults with Developmental Disabilities (AADS) [79]. This instrument assesses prodromal behaviour modifications and deficits associated with dementia in people with ID—such as agitation, stereotypical behaviour, anxiety, or inactivity. The adaptive behaviour dementia questionnaire (ABDQ) is another instrument specifically developed to assess behaviour changes in those with ID and dementia [6]. The ABDQ was used in two cross-sectional studies [80,81].

#### 4.2.4. Limitations

There are some limitations to this review. There is a lack of findings from studies published in other languages. For instance, De Vreese et al. (2011) [62] carried out an Italian adaptation of the AADS (AADS-I) that displays good psychometric properties and satisfactory interrater reliability for the six subscales (coefficients from 0.67 to 0.79). A further limitation is the lack of studies found in grey literature and open science databases; while only including papers from peer-reviewed journals helps to ensure the quality of included studies is high, it also limits a large amount of research which may provide additional insights.

Another limitation is the lack of psychometric data for some of the instruments used. Although we aimed to create a review to help clinicians and researchers to find the most suitable instrument, many studies did not provide psychometric properties based on their samples, and we considered it inappropriate to use secondary sources, such as tests and batteries handbooks, as they do not reflect characteristics of the current samples.

As with any diagnostic assessment, we recommend following practical medical guidelines with multiple diagnostic approaches assessing cognitive, behavioural, and independent functioning. The use of informant and self-report instruments alongside medical examinations, neuroimaging techniques, and genetic and biological measures of various types of dementia is also recommended [82].

We found no overlap between measures used across studies, with no measure included in more than one study. The use of the same instruments in different languages would favour cross-cultural comparisons. This is illustrative of the lack of standardised measures for assessing cognitive decline in those with other ID and highlights the need for an accepted, recommended measure to allow synthesis across different studies.

This systematic review could not examine neuropsychological assessment in different stages of dementia due to the nature of the articles selected. There is no consensus regarding dementia stages in people with ID and discrepancies with the general population are observed [8,63,83]. This reinforces the need for longitudinal studies to investigate cognitive changes in DS and other ID. Some studies [18,61] show promising examples of the benefit of this approach, as they use baseline and longitudinal data to support and explore factors related to cognitive decline.

## 5. Conclusions

In conclusion, there is a multitude of instruments being used to screen for cognitive changes associated with dementia in those with ID. This review highlights the variation between measures used across studies and illustrates the need for unified, standardised measures to allow for the synthesis of results in research and greater consistency of diagnosis in clinical practice. Contrasting cross-sectional and longitudinal studies, we recommend the use of specifically designed instruments, such as the DLD [14] and the CAMCOG-DS [67], to assess cognitive functioning and behaviour changes related to ID and dementia. The use of measures designed for the general population should be avoided due to their lack of sensitivity in differentiating between those with and without dementia. Evidence supports the DLD as a promising informant-based screening tool for the diagnosis of dementia, since it covers both cognitive and behavioural symptoms [38,84]. We stress, however, that the DLD is not an instrument for a clear-cut diagnosis, but rather a good screening instrument for follow up assessment which is reliable when used routinely in combination with other objective measures such as, for example, CAMCOG-DS.

## Figures and Tables

**Figure 1 brainsci-10-00848-f001:**
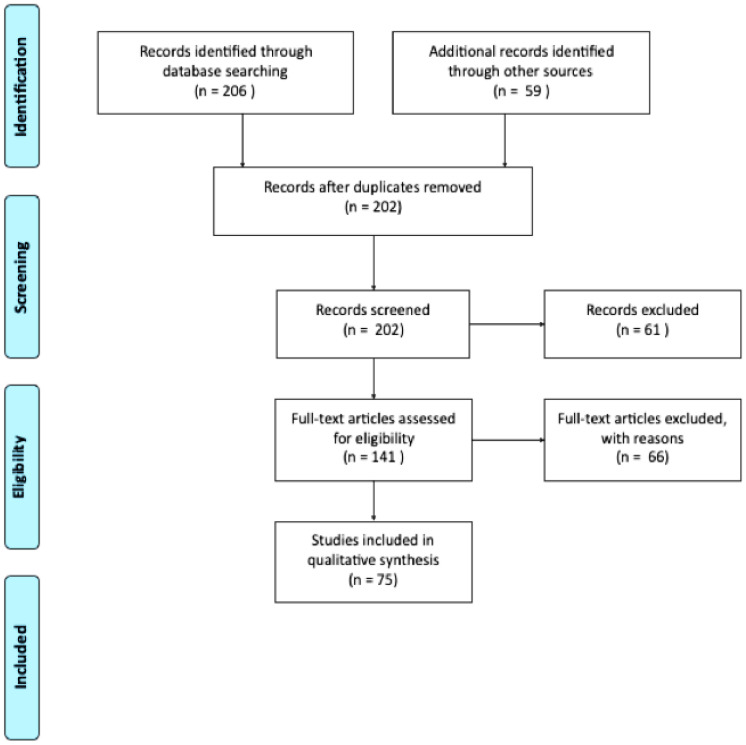
Preferred reporting items for systematic review and meta-analysis (PRISMA) flow chart concerning study retrieval and selection.

**Table 1 brainsci-10-00848-t001:** Demographics of included individuals in the eligible studies.

Cross-Sectional Studies		Longitudinal Studies	
Down Syndrome	2776	Down Syndrome	1874
Other ID	1231	Other ID	531
Male	1396	Male	110
Female	1143	Female	450
Missing Data	1482	Missing Data	1284
Total	4007	Total	2405

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
