# Peer review of "Screening of Cognitive Changes in Adults with Intellectual Disabilities: A Systematic Review"

_brainsci, 2020, doi:10.3390/brainsci10110848_

Round 1
Reviewer 1 Report
This is a well presented and insightful article highlighting the utility of the Screening and assessment of cognitive changes in adults with Intellectual Disabilities. The analysis and findings are thorough and robustly undertaken. Strengths and limitations are clearly presented.
I would only like to point out that another recent article published on 3 July 2020 and which has partly different objectives (Krinsky-McHale et al., Promising outcome measures of early Alzheimer's dementia in adults with Down syndrome DOI: 10.1002/dad2.12044) also reports that DLD is included in the measures that "provide essential tools to inform current clinical diagnosis" especially for people with suspected dementia
Author Response
Dear reviewer, we do appreciate your appraisal of our work! We included the new article in our review. By the way, thanks for pointing this.
Reviewer 2 Report
The organization of the manuscript remains difficult to follow. If this is a review of measures, the manuscript should be organized as such, as opposed to being organized by longitudinal vs. cross-sectional study in which they were used.
Author Response
Dear reviewer,
This manuscript is a resubmission of an earlier submission. The following is a list of the peer review reports and author responses from that submission.
Round 1
Reviewer 1 Report
It seems to me an interesting and well-constructed study with rigorous criteria, I also appreciated because it helps to guide researchers, but also health service providers to use standardized tools, which we have an evident replicability even in multicultural contexts and above all with highlighted psychometric properties .
The comparison criteria and the search of the databases seem to me well explained and clear as well as the exclusion criteria.
I do not detect particular criticalities and the references seem to me up to date with the current research.
Reviewer 2 Report
Topic to include DS and all with ID is too broad to be clinically useful.
Reviewer 3 Report
- The organization of the paper is difficult to follow. I am not sure that there is a need to separate measures out by whether the study reviewed was longitudinal or cross-sectional, when in fact the same measure is often used in both types of studies. If anything, it makes more sense to separate measures by informant- vs. self-report.
- It would make more sense to organize the discussion by measure as opposed to discussing the same measure in more than one section. The organization by study methodology and whether the study included people with DS vs. people with other ID is difficult to follow.
- There needs to be a deeper discussion of the implications of this review. What does this add to the literature?
- The focus of your study is on the use of a measure, yet your tables do not describe the validity of the dementia screener or cognitive test, but rather describe broader study outcomes. For example, in your table, for study 48 the outcomes is: 56% of participants preferred facial
pictograms scales over drawn face
stimuli. How is this relevant to your overarching study purpose/discussion of tools to assess dementia in this population?